# Adaptability and Resilience of Academic Radiation Oncology Personnel and Procedures during COVID-19 Pandemic

**DOI:** 10.3390/ijerph18105095

**Published:** 2021-05-12

**Authors:** Indra J. Das, John A. Kalapurakal, Jonathan B. Strauss, Brian R. Zawislak, Mahesh Gopalakrishnan, Amishi Bajaj, Bharat B. Mittal

**Affiliations:** Department of Radiation Oncology, Northwestern Memorial Hospital, Northwestern University Feinberg School of Medicine, 251 East Huron Street, Galter Pavilion, LC-178, Chicago, IL 60611, USA; John.Kalapurakal@nm.org (J.A.K.); jonathan.strauss@nm.org (J.B.S.); Brian.Zawislak@nm.org (B.R.Z.); Mgopalak@nm.org (M.G.); Amishi.Bajaj@nm.org (A.B.); Bharat.mittal@nm.org (B.B.M.)

**Keywords:** COVID-19, radiation oncology, operation, survey, adaptability, resilience

## Abstract

Background: A comprehensive response to the unprecedented SARS-CoV-2 (COVID-19) challenges for public health and its impact on radiation oncology patients and personnel for resilience and adaptability is presented. Methods: The general recommendations included working remotely when feasible, implementation of screening/safety and personal protective equipment (PPE) guidelines, social distancing, regular cleaning of treatment environment, and testing for high-risk patients/procedures. All teaching conferences, tumor boards, and weekly chart rounds were conducted using a virtual platform. Additionally, specific recommendations were given to each section to ensure proper patient treatments. The impact of these measures, especially adaptability and resilience, were evaluated through specific questionnaire surveys. Results: These comprehensive COVID-19-related measures resulted in most staff expressing a consistent level of satisfaction in regard to personal safety, maintaining a safe work environment, continuing quality patient care, and continuing educational activities during the pandemic. There was a significant reduction in patient treatments and on-site patient visits with an appreciable increase in the number of telemedicine e-visits. Conclusions: Survey results demonstrated substantial adaptability and resilience, including in the rapid recovery of departmental activities during the reactivation phase. In the event of a future public health emergency, the measures implemented may be adopted with good outcomes by radiation oncology departments across the globe.

## 1. Introduction

About a century ago, the influenza pandemic of 1918 (“the Spanish Flu”) infected nearly half a billion and killed nearly 50 million people worldwide [1]. A similar outbreak of SARS-CoV-2 (COVID-19) was recognized in 26 December 2019 and has since rapidly spread to become a pandemic posing critical challenges for public health, the global economy, and clinical medicine [2,3]. As of March 2021, COVID-19 has infected over 126 million people worldwide and resulted in 2.8 million deaths (COVID-19 Statistics, 2021). Unfortunately, hospitals worldwide were not immediately prepared to cope with the impact of this pandemic, scrambling to prioritize resources for identifying, testing, and managing critically ill COVID-19 patients [4]. The disease spreads rapidly to airways with inflammation of the respiratory system within a few days with the appearance of pneumonia and sepsis. Chest imaging using X-rays and CT scans shows patchy or diffuse asymmetric airspace opacities, a distinct ground glass patchy pattern in lungs [5,6,7]. A few medical options to treat patients had minimal gain [8]. Most elective procedures in the hospitals were suspended to support the pandemic health crisis.

The Unites States has been the hardest hit with COVID-19 cases and death. Every hospital has put on hold the wards to manage the most infective disease, thus, limiting treatment to other diseases. Early signs indicate that the death rate is much higher in patients with smoking, obesity, diabetic, respiratory disease, and cancer. Thus, radiation oncology was hard hit to limit viral exposure to the suffering cancer patients. The radiation oncology community tried to treat cancer patients with variable precautions, as listed in recent literature with altered practice as necessitated by local needs [9,10]. To limit exposure to staff, rotation schemes among MDs, nurses, and administrative and front desk staff have been implemented. On the other hand, all treatment planning were remotely performed with skeletal staff in the department. Similarly, physicists were rotated to evaluate QA, chart checks, machine status, and to deal with special procedures such as gamma knife, HDR, TBI, etc. Maintaining the performance and calibration of the machine was often an after work job where viral exposure was a minimum with keeping distance. Most of the work in radiation oncology was redistributed to work from home if possible during the COVID-19 pandemic from February to May 2020. 

This drastic decision to limit staffing in the department is contrary to our thoughts, since safety is directly proportional to direct intervention and error reporting [11]. Many of the incidences in radiation oncology have provided feedback for the culture of safety advocated by various societies [12,13,14,15]. However, during COVID-19, full staffing could not be possible, and hence, an evaluation of various parameters, safety, satisfaction, and treatment quality needed to be evaluated.

The COVID-19 pandemic has affected all aspects of life. Nations and organizations have largely adapted policies and procedures to cope with and adjust to the challenges posed by the virus without much planning. Without a talisman and a lack of effective treatments, infected patients were managed with supportive measures, and stay-at-home orders and closure of all non-essential businesses were required for society at large. Social distancing and the use of personal protective equipment (PPE) were recommended [4,16]. Essential healthcare services such as urgent care centers and hospitals remained open, while non-emergent surgical and diagnostic procedures were canceled. 

Surgical procedures provide patients for radiation therapy, but in the face of COVID-19, most elective surgeries were curtailed with guidelines provided by the surgical society along with psyche to surgeon, described by Cobianchi et al. [17]. Surgeons were forced to shift from patient-centric ethics to public health ethics. This forced them to alter surgical procedures, thus, limiting the pipeline for radiation patients. Additionally in most part of the world, challenging issues were the scarcity of PPE, rationing of ventilators, and ICU beds for elective surgery [17,18]. Cancer treatments were postponed when it was deemed not safe when weighed against the risk of contracting COVID-19 infection [9,19]. Radiation oncology communities attempted to treat cancer patients with variable precautions with altered practice as necessitated by local needs [9,10]. There has been a suggestion of low-dose radiation to the whole lungs for COVID-19 treatment [20], but the consequences and opposition of radiation are also debated [21].

Barcellini et al. [22] provided a new normal with guidelines for social distancing, infection presentation, and options for radiation treatment during COVID-19. ASTRO/ESTRO provided consensus guidelines to address the head and neck radiation therapy treatment [23]. They recommended the use of hypofractionation, delay of treatment until the patient is proven negative for COVID-19, and no interruption in treatment if radiation treatment has started. The multi-institutional survey conducted by the European group illustrated that 46.7% centers were affected by Covid-19 but continued with radiotherapy treatment in the face of the COVID-19 pandemic [24]. 

Most of the COVID-19 literature deals with managing patients, and none of them provide meaningful data on the state of staff. In this context, our study describes the comprehensive response undertaken by radiation oncology staff in the face of COVID-19 at an academic institution in the United States, seeks to understand the impact on personnel and procedures, and demonstrates the adaptability and resilience noted following implementation of mitigating measures on the face of the COVID-19 pandemic.

## 2. Materials and Methods

### 2.1. Mitigating Procedures

The comprehensive response detailed below was developed jointly by the hospital administration in conjunction with infection prevention department called the COVID-19 task force and radiation oncology leadership. These teams held regular teleconferences initially daily and then weekly to manage the changing crisis with proper recommendations.

### 2.2. General Recommendations

Hospital infection control policies were strictly followed;Everyone entering the department was monitored for temperature and screened for symptoms such as fever, cough, or chest pain. If febrile or symptomatic, employees were directed to receive COVID-19 testing;Staff treating COVID-19-positive patients were required to wear full PPE, gloves, mask, gowns, goggles, or face shields);Everyone was required to wear masks and gloves and to wash their hands with soap/hand sanitizers before and after patient contact;Visitors were allowed in the department only when needed;All treatment devices and surfaces were cleaned with disinfectant wipes after each treatment;Exam room doors remained open for contactless operation and air exchange;Consultations and follow-ups were held using telemedicine;COVID-19-positive patients were treated at the end of the day on a designated machine;A two-week quarantine period was mandated when a member living in the staff/patient’s home tested positive;Patients were tested negative for special procedures (brachytherapy, gamma knife, and SBRT);Children requiring sedation for irradiation were tested weekly;Laptop computers and database access were provided to staff for remote work;All teaching conferences, tumor boards, and weekly chart rounds were virtually conducted using teleconferencing.

### 2.3. Physicians

Patients were triaged into three groups; (1) patients with radiation emergencies, (2) patients whose treatment could not be postponed, and (3) patients with cancers where treatment could be safely postponed. In general, treatment was only given for groups 1 and 2;Shorter treatment courses were used when feasible [9,10,19];Only one on-call physician was in the department daily; others were in the clinic only once a week for patient’s on-treatment visits.

### 2.4. Nursing

Two nurses were required to be in the clinic;Nurses were responsible for maintaining a clean and contamination-free clinic and observing;Social distancing among patients was enforced.

### 2.5. Physicists

At least two physicists were on site for 2–3 days each week;Additional staffing were required on site for special procedures (two physicists each for brachytherapy and gamma knife and one for the SBRT);Treatment plans and chart checks were performed by off-site physicists;Calibration machine maintenance and QA were performed after hours.

### 2.6. Dosimetrists

Two dosimetrists were available on site daily;Every dosimetrist was provided a laptop computer to facilitate remote work;Communications were maintained as needed via email, phone, and teleconferencing.

### 2.7. Radiation Therapists

A two-week rotation with 3 days/week for treatment was adopted;When patient volumes decreased to 50%, they were placed in the hospital’s “labor pool” to allow them to work and be paid off;Instructed to remain in their assigned rooms to minimize cross exposure;After each treatment, they were required to disinfect every treatment device and surface.

### 2.8. Residents and Students

Students were instructed to stay at home unless needed;One on-call resident was required to be in the clinic each week, while the others were allowed to work from home;All clinical and educational activities were performed remotely.

### 2.9. Laboratory Research

Laboratory research was halted during the COVID-19 period except data analyses, computational work, maintenance of cell lines, and freezer stocks;Research staff were instructed to stay at home;Computational research support was available to staff and faculty.

### 2.10. Impact of Departmental Response

For this report, the pandemic timelines were pre-COVID-19 (29 February 2020), COVID-19 (1 March 2020 until 10 May 2020), and the reactivation phase (11 May 2020 until 29 May 2020). The impact of comprehensive measures on personnel and procedures was determined by the following:Questionnaire survey response from all staff based on Institutional Review Board (IRB) exempt status;Census of departmental activities (consultations, treatments, and attendance during teaching and clinical conferences);COVID-19 infection rates among staff and patients;Rates of normalization of activities during the reactivation phase.

## 3. Results

### 3.1. Questionnaire Response

The responses to a section-specific questionnaire sent to staff are summarized in Figure 1. The compliance to the survey was 100% (*n* = 73), as every clinical member of the department participated in the survey privately and remotely. The scores ranged from 0–10, with 0 indicating the least and 10 the highest satisfaction. In majority of the group the error bar was rather large, indicating large variations in each group. Additionally, the satisfaction was limited and varied among the group. The lowest was in therapists who were dealing with the COVID-19 crisis daily. Those groups who were working remotely had better performance. 

While questionnaires were section-specific, three questions (adaptability of COVID-19, work-life balance, and satisfaction with management) were common to the entire department. These were analyzed with results shown in Figure 2, indicating a consistent level of satisfaction with the proposed measures aimed at ensuring personal safety, maintaining a safe work environment, continuing quality patient care, and providing educational activities during the pandemic. The error bars represent the variation (1 standard deviation) among the individual questionnaire in a section. The data from this survey and institutional survey conducted before pandemic are nearly identical, indicating that satisfaction with the management and patient care were not impacted due to pandemic, which indicates a high degree of resilience in the staff.

### 3.2. Census of Departmental Activities

The census of patient consultations, simulations, and treatments (Figure 3) shows the negative impact of the pandemic on clinical activities. The number of patients treated dropped significantly (±20%) during the COVID-19 period, and the on-site consults and follow-ups fell rapidly to nearly 1/3 during the COVID-19 phase. However, this was compensated by a significant increase in the number of telemedicine e-visits. The use of telemedicine has enabled consistent levels of participation by staff and residents at pre-COVID-19 levels for all educational conferences (journal clubs, didactic lectures) and weekly chart rounds.

### 3.3. COVID-19 Infection Rates

During this reporting period, none of the staff or students have developed symptoms or signs of COVID-19 infection. Six patients who tested positive for COVID-19 were treated according to standard protocols. None of these patients died. Two patients required radiation treatment while they were COVID-19-positive. During the reactivation phase, a COVID-19 surge in the USA took place where we had 8% positive patients treated without any staff being infected.

### 3.4. Census of Activities during the Reactivation Phase

The high morale and resilience of our departmental clinical staff and the support of the hospital authorities resulted in the rapid recovery of departmental activities during the reactivation phase as indicated in Figure 3.

## 4. Discussions

Most of the COVID-19 literature was published mainly for patient care. A few dealt with specific areas such as the impact on surgeon [17,18]. There are specific guidelines for radiation oncology patient management [10,19,22,23,25]. Our study was mainly to inspect the impact of institutional COVID-19 guidelines, precautions, infection control, and patient treatment on every section of the department. 

In the face of a global pandemic with a virus lacking effective treatment, it is natural for clinical staff who are concerned for their own and their family’s wellbeing to also be fearful for having to choose between their clinical obligations and the stay-at-home orders issued by state governments. The comprehensive measures described in this report were implemented with regular feedback and collaboration among all staff to effectively deal with the pandemic and mitigate its impact on personnel and procedures. The questionnaire responses indicated that there was general satisfaction with the overall goal of ensuring personal safety and providing a good work environment very similar to the data provided by Reuter-Oppermann et al. [24]. There was a significant reduction in the number of patients treated during the COVID-19 phase, as patients were triaged depending on the urgency of their diagnosis as described earlier [9,10,19]. The use of telemedicine was immensely helpful in enabling our patients to continue to receive medical advice. Additionally, it was critical in mitigating the potentially negative impact on resident education and other departmental conferences [26,27]. 

The closure of all laboratories and cessation of basic research has adversely affected animal research. The low rates of symptomatic infections among our patients and staff supports the value of social distancing, use of PPE, and maintaining a contamination-free clinical environment. A significant weakness of this report is the lack of periodic and comprehensive testing for COVID-19 infection for staff and patients. This was due to the national shortage of testing kits and the strategic decision to direct such resources to symptomatic staff and patients [28].

This study is limited to an academic department with multiple sections. However, one can extrapolate this data for a community practices. Another caveat is the location of our center, which is in down town of a very large metropolis, and our experience may not directly reflect that of the rural centers. 

Post-pandemic studies will provide valuable guidelines on the psyche of staff and its long-term effect on every aspect of patient care. Additionally, postponement or delay and altered fractionation will provide valuable guidelines for future treatment regimens and outcomes related to radiation treatment. 

## 5. Conclusions

This study describes the lessons learned from a comprehensive strategy for all staff and procedures in an academic radiation oncology department during the COVID-19 pandemic. The recommendations were successful as shown by the survey in maintaining staff morale and safety, delivering treatment safely to patients, and maintaining the quality of resident education during the pandemic. With proper precautions, radiation oncology patients could be successfully treated as shown by the survey indicating adequate resilience and adaptability in the face of COVID-19. These measures can be easily adopted by any radiation therapy department with good outcomes in the event of a second wave or future pandemics.

## Figures and Tables

**Figure 1 ijerph-18-05095-f001:**
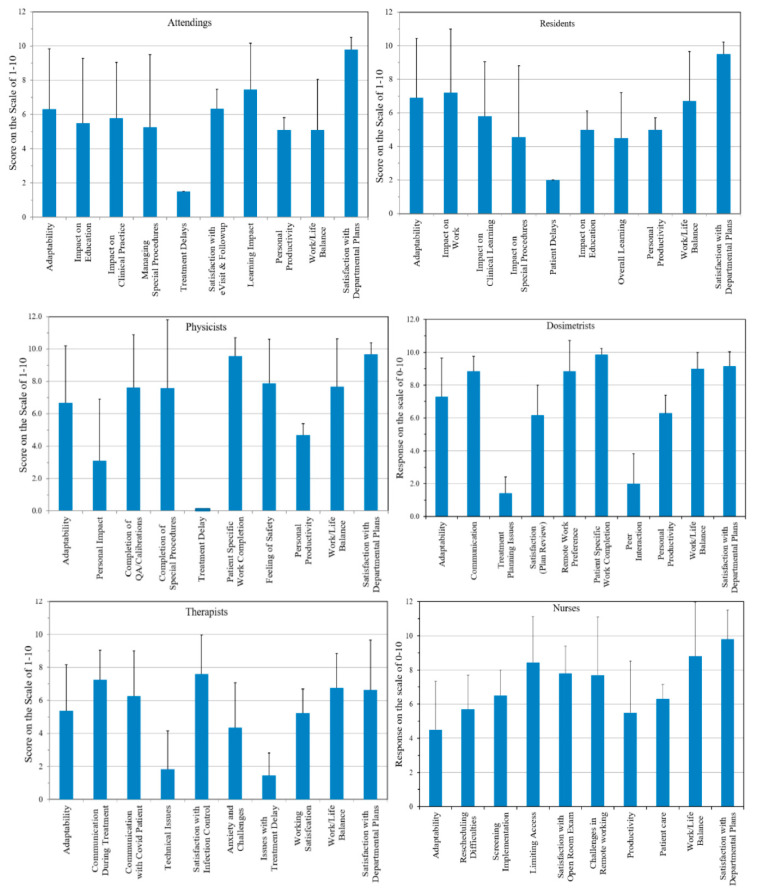
Questionnaire responses from each section of the department. Questions in each group are not identical.

**Figure 2 ijerph-18-05095-f002:**
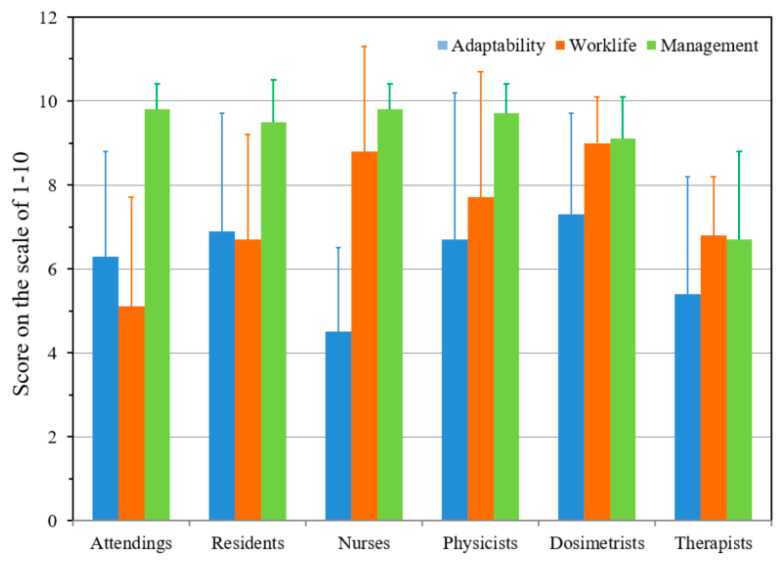
Analysis of the three common questions for all staff members. Please note the adaptability is variable in each group depending on the contacts with patients.

**Figure 3 ijerph-18-05095-f003:**
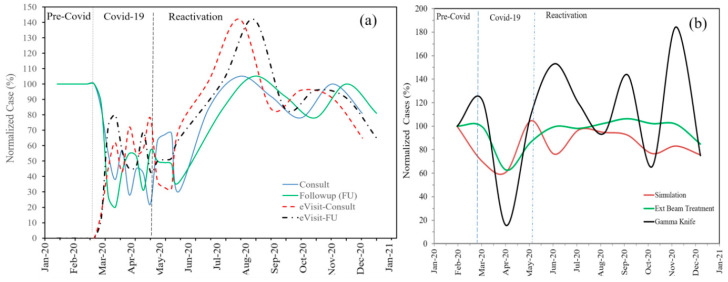
(**a**) Pattern of consults, follow-ups, and respective e-visits using telemedicine in 3 time scales; pre-COVID, COVID-19, and post-COVID-19 (reactivation—resumption of normal operation), (**b**) census of patient simulations, treatments, and gamma knife. Note that during COVID-19 most activities were significantly reduced to nearly 20% of the normal load.

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
