# Peer review of "Adaptability and Resilience of Academic Radiation Oncology Personnel and Procedures during COVID-19 Pandemic"

_ijerph, 2021, doi:10.3390/ijerph18105095_

Round 1

Reviewer 1 Report

Thank you for the opportunity to review this manuscript.

I think this analysis nicely tries to summarize the pandemic action in a Radiation Oncology Department in the midst of the pandemic.

The results in Figure 3a are interesting but not easy to read.

The discussion should be implemented considering also the other experiences of the oncological department reported in the recent literature.

Moreover, considering the aim of this paper and its results, the discussion of the results should be implemented and compared with the anti-fragile literature; indeed in recent months, the procedure related to patient-centered ethics have been replaced by public health ethics with new strategies for a re-arrangement of the routine practice in anti-fragile perspective.

Consider:

  • Angelos P. Surgeons, Ethics, and COVID-19: Early Lessons Learned. J Am Coll Surg. 2020 Jun;230(6):1119-1120. doi: 10.1016/j.jamcollsurg.2020.03.028
  • Barcellini A, Filippi AR, Dal Mas F, Cobianchi L, Corvò R, Price P, Orlandi E. To a new normal in radiation oncology: looking back and planning forward. Tumori. 2020 Dec;106(6):440-444. doi:10.1177/0300891620962197
  • Cobianchi L, Dal Mas F, Peloso A, Pugliese L, Massaro M, Bagnoli C, Angelos P. Planning the Full Recovery Phase: An Antifragile Perspective on Surgery After COVID-19. Ann Surg. 2020 Dec;272(6):e296-e299. doi: 10.1097/SLA.0000000000004489

Reviewer 2 Report

Authors should be commended for researching practical components regarding application of RTx. in Covid-19 pandemic. However, to be honest, this article feels like more a social one rather than a scientific article.

1) In line 88, author mentioned that "developed jointly"; however, more formal and detailed information about how such guidelines were generated. By whom, via which methods.. etc.

2) Actual questionnaire or related information should be provided.

3) Please be more specific about the conclusion of main text and abstract. What message do the authors want to convey through this study? In the manuscript as it is, this is too vague.

There are several typo-errors; please check when re-submit.

Round 2

Reviewer 2 Report

This study has average quality overall.

I have no strong objection for rejection of acceptance.